# Plaque Characteristics Derived from Intravascular Optical Coherence Tomography That Predict Cardiovascular Death

**DOI:** 10.3390/bioengineering11080843

**Published:** 2024-08-19

**Authors:** Juhwan Lee, Yazan Gharaibeh, Vladislav N. Zimin, Justin N. Kim, Neda S. Hassani, Luis A. P. Dallan, Gabriel T. R. Pereira, Mohamed H. E. Makhlouf, Ammar Hoori, David L. Wilson

**Affiliations:** 1Department of Biomedical Engineering, Case Western Reserve University, Cleveland, OH 44106, USA; jxl1982@case.edu (J.L.); jnk50@case.edu (J.N.K.); aoh11@case.edu (A.H.); 2Department of Biomedical Engineering, Faculty of Engineering, The Hashemite University, Zarqa 13133, Jordan; yazan@hu.edu.jo; 3Brookdale University Hospital Medical Center, 1 Brookdale Plaza, Brooklyn, NY 11212, USA; vlzimin@gmail.com; 4Harrington Heart and Vascular Institute, University Hospitals Cleveland Medical Center, Cleveland, OH 44106, USA; neda.shafiabadihassani@uhhospitals.org (N.S.H.); luisaugusto.palmadallan@uhhospitals.org (L.A.P.D.); gabriel.tensolrodriguespereira@uhhospitals.org (G.T.R.P.); mohamed.makhlouf@uhhospitals.org (M.H.E.M.); 5Department of Radiology, Case Western Reserve University, Cleveland, OH 44106, USA

**Keywords:** intravascular optical coherence tomography, cardiovascular death, plaque characteristics, fibrous cap surface area, OCTOPUS

## Abstract

This study aimed to investigate whether plaque characteristics derived from intravascular optical coherence tomography (IVOCT) could predict a long-term cardiovascular (CV) death. This study was a single-center, retrospective study on 104 patients who had undergone IVOCT-guided percutaneous coronary intervention. Plaque characterization was performed using Optical Coherence TOmography PlaqUe and Stent (OCTOPUS) software developed by our group. A total of 31 plaque features, including lesion length, lumen, calcium, fibrous cap (FC), and vulnerable plaque features (e.g., microchannel), were computed from the baseline IVOCT images. The discriminatory power for predicting CV death was determined using univariate/multivariate logistic regressions. Of 104 patients, CV death was identified in 24 patients (23.1%). Univariate logistic regression revealed that lesion length, calcium angle, calcium thickness, FC angle, FC area, and FC surface area were significantly associated with CV death (*p* < 0.05). In the multivariate logistic analysis, only the FC surface area (OR 2.38, CI 0.98–5.83, *p* < 0.05) was identified as a significant determinant for CV death, highlighting the importance of the 3D lesion analysis. The AUC of FC surface area for predicting CV death was 0.851 (95% CI 0.800–0.927, *p* < 0.05). Patients with CV death had distinct plaque characteristics (i.e., large FC surface area) in IVOCT. Studies such as this one might someday lead to recommendations for pharmaceutical and interventional approaches.

## 1. Introduction

Coronary artery disease (CAD) remains a significant cause of morbidity and mortality worldwide [1]. Due to the widespread prevalence of atherosclerosis [2], there is an increasing interest in accurately assessing disease aggressiveness by predicting major adverse cardiovascular events (MACE) through medical imaging. Various imaging modalities, such as coronary computed tomography (CCTA) and intravascular optical coherence tomography (IVOCT), have been utilized to visualize plaques in coronary vessels and characterize CAD. In CCTA images, several high-risk features, such as positive remodeling, low-attenuation plaque, napkin-ring sign, and spotty calcification, have been identified, with their association with MACE serving as a key determinant [3,4,5,6]. Recent research has also emphasized the assessment of pericoronary fat in CCTA images, including radiomic evaluations, which again leverage their association with MACE to support their clinical relevance [5,7,8,9,10,11,12]. Using CT calcium score images, MACE can be predicted using the Agatston score derived from calcifications [13,14,15].

None of the currently available non-invasive imaging modalities can match the exceptional resolution and contrast achieved with IVOCT images [16]. Notably, IVOCT provides superior resolution (axial: 10 μm, lateral: 20–40 μm), allowing for the precise identification of thin cap fibroatheroma (TCFA). Through IVOCT imaging, microscopic characteristics of plaque, including TCFA, macrophage infiltration, cholesterol crystal presence, and microchannels, can be quantitatively assessed, as validated by histological studies [17,18,19,20,21,22,23,24,25,26,27,28,29,30,31,32,33,34,35,36]. Histopathological investigations have also shown a strong association between the pathogenesis of most acute coronary events (such as plaque rupture and myocardial infarction) and the presence of microcalcification, TCFA, and large lipid-rich necrotic cores [37,38,39]. These plaques are also often characterized by intraplaque hemorrhage and inflammation, both of which are strongly associated with plaque progression [40]. Extensive ex vivo and in vitro studies have further supported the correlation between macrophage infiltration and vulnerable plaque characteristics through histopathological evidence [37,38,41,42,43]. As IVOCT provides a promising, unique perspective on coronary plaque, there is a strong rationale to relate IVOCT findings to long-term cardiovascular risk, specifically in terms of predicting future adverse events.

There have been relatively few studies analyzing IVOCT images to predict adverse events [44,45,46,47,48]. In a landmark study involving 1474 patients, Xing et al. focused on the role of lipid-rich plaque within the non-culprit region of target vessels in predicting MACE [44]. Using conditional logistic regression, they found a significant correlation between the presence of lipid-rich plaques and increased MACE risk (risk ratio: 2.1, 95% CI: 1.1–4.0, *p* = 0.036). Individuals who experienced MACE had longer lipid lengths (*p* < 0.002) and broader maximal lipid arcs (*p* = 0.023) compared to those without MACE. Prati et al. investigated the predictive power of various high-risk plaque attributes captured in IVOCT images as predictors of future adverse events [45]. Their 1-year follow-up findings indicated that a minimum lumen area < 3.5 mm^2^ (hazard ratio [HR] 2.1, 95% confidence interval [CI] 1.1–4.0), fibrous cap (FC) thickness (HR 4.7, 95% CI 2.4–9.0), circumferential lipid arc > 180° (HR 2.4, 95% 1.2–4.8), and the presence of macrophages (HR 2.7, 95% CI 1.2–6.1) significantly elevated the risk of future adverse events. Montone et al. examined the relationship between macrophage infiltration at the culprit site with plaque erosion and MACE incidence [46]. They found a significantly higher occurrence of MACE among patients with plaque erosion and macrophage infiltration compared to patients without macrophage infiltration (21.6% vs. 5.9%, *p* = 0.008). Zhao et al. assessed the prognostic value of plaque characteristics and residual syntax score for predicting MACE [47]. Their results revealed a significant association between the presence of TCFA and higher residual syntax scores with the occurrence of MACE. In another study, Kim et al. analyzed post-stenting IVOCT findings to identify predictors of adverse outcomes [48]. They reported that a smaller minimal stent area and malapposition with a total malapposition volume ≥ 7.0 mm^3^ were found to be independent predictors of device-oriented clinical endpoints and major safety events, respectively.

Despite these promising findings, there is a lack of comprehensive analysis of plaque characteristics, particularly the FC, derived from IVOCT imaging as predictors of future adverse events. We hypothesize that plaque characteristics derived from IVOCT are related to the presence of cardiovascular (CV) death. Specifically, we aim to identify relevant features in pre-stent IVOCT images that exhibit an association with CV death. This analysis will not only reveal high-risk features extracted from IVOCT but will also enable subsequent correlative investigations with CCTA to identify novel high-risk CCTA features based on their association with IVOCT findings. In this context, we have identified a correlation between specific radiomic features of pericoronary fat in CCTA images and microscopic IVOCT characteristics, including TCFA and microchannels [49]. It is important to note that high-risk features identified through their association with MACE may differ from those associated with risk in a specific lesion, as identified solely by CCTA.

In this study, we investigated the association between plaque characteristics observed in IVOCT imaging and the occurrence of CV death. Using Optical Coherence TOmography PlaqUe and Stent (OCTOPUS) software [50] on IVOCT images, we segmented plaques into constituent parts (e.g., calcium and FC), extracted IVOCT plaque features (e.g., calcium thickness, calcium angle, and FC surface area), and determined their discriminatory power for predicting CV death. Notably, this analysis specifically focused on microscopic features that are exclusively visible through IVOCT, such as FC and microchannel components.

## 2. Materials and Methods

### 2.1. Study Population

We retrospectively reviewed 805 patients and enrolled 104 patients with coronary artery disease who had undergone clinically indicated invasive coronary X-ray angiography and IVOCT-guided percutaneous coronary intervention (PCI) at University Hospitals Cleveland Medical Center in Cleveland, Ohio, USA, between 8 February 2013, and 30 November 2019. These patients were primarily selected for another study [51], which required paired pre- and post-stenting IVOCT pullbacks, leading to the exclusion of a large number of patients. We included patients who had a culprit lesion identified through coronary X-ray angiography. The exclusion criteria included poor-quality images, ostial lesions, inability to cross the lesions with the OCT catheter due to the tortuosity and/or occluding thrombus, bypass graft stenosis, in-stent restenosis, and chronic total occlusions. Data collection occurred from September 2019 to May 2021, and the study data were subsequently analyzed in 2022 and 2023. This study was conducted in compliance with the Declaration of Helsinki and received approval from the Institutional Review Board of University Hospitals Cleveland Medical Center, Cleveland, Ohio, USA (STUDY20190821). The requirement for individual informed consent was waived as all data were fully anonymized, with no identifiable personal health information.

### 2.2. IVOCT Imaging and Plaque Characterization

Invasive coronary angiography was performed using 6–7 Fr catheters via radial or femoral access, following the administration of 250 µg of intracoronary nitroglycerine. The resulting coronary angiogram was analyzed using QAngio^®^ software (v7.3, Medis, Leiden, The Netherlands). IVOCT-guided PCI was conducted employing conventional techniques. During the procedure, the interventional cardiologist exercised discretion in selecting stenting variables such as stent length and diameter. Only drug-eluting stents were utilized in this study. IVOCT images were acquired using the C7XR FD-OCT imaging system (Abbott Vascular, Santa Clara, CA, USA) following the administration of nitroglycerin (100–200 g). To reach the lesion of interest, a 2.7-Fr OCT catheter (Dragonfly OPTIS, Abbott Vascular, Santa Clara, CA, USA) was advanced over a conventional guidewire, with the catheter position verified through invasive coronary angiography. Non-diluted iodine contrast (ISOVUE-370, iopamidol injection, 370 mg iodine/mL; Bracco Diagnostics Inc., Princeton, NJ, USA) was used to achieve blood clearance. Imaging pullback was then performed at a frame rate of 180 fps, a pullback speed of 36 mm/s, and an axial resolution of approximately 20 µm. The acquired images were de-identified and forwarded to the Cardiovascular Imaging Core Laboratory for independent offline analysis.

Plaque and vessel analysis was conducted semi-automatically using the OCTOPUS software, which was previously developed and validated by our research group [50]. Briefly, OCTOPUS automatically segmented the lumen, lipid, calcification, and microchannels using a modified version of DeepLab v3+ deep-learning model [52]. In cases of detected lipidic plaque, the FC plaque regions were identified using dynamic programming, as previously proposed by our group [53]. Additionally, the presence of vulnerable plaques, including macrophage infiltration, cholesterol crystals, layered plaques, and calcium nodules, was manually assessed using the OCTOPUS software. If necessary, manual editing of the results was performed with an interactive editing tool following the definitions provided in the “consensus document” [54]. Detailed algorithm descriptions, extensive assessment, and promising results are provided elsewhere [55,56,57,58,59].

### 2.3. IVOCT Feature Selection

We analyzed 31 IVOCT features from baseline IVOCT images taken prior to stenting to predict the occurrence of CV death. The features were automatically computed using OCTOPUS software [50], with the exception of the vulnerable plaque features. Lesion length was defined as the length of the vessel segment where the stent was deployed. Lumen features included minimum and average lumen area, as well as minimum and average lumen diameter. Calcium features included maximum and minimum calcium angle, thickness, and depth. As shown in Figure 1, FC features were evaluated across four levels of FC thickness (1: thickness ≤ 65 µm, 2: 65 µm < thickness < 150 µm, 3: thickness ≥ 150 µm, and T: total) and included maximum and minimum FC angle, thickness, area, surface area, and burden. For example, FC surface area-1 represented the surface area of FC regions with a thickness ≤ 65 µm, while FC surface area-T represented the total surface area of FC regions, regardless of thickness. FC angle, thickness, and area were calculated from each IVOCT image frame, while FC surface area and burden were computed for the entire lesion. FC surface area was defined as the total area covered by the FC on the surface of the vessel lumen, as visualized in the en face view (*θ*,*z*), while FC burden was calculated as the ratio of FC area to the surface area of the vessel lumen. The angles of calcified and FC plaques were determined based on the extent of plaques relative to the center of mass of the lumen. Vulnerable plaque features included the presence of microchannels, macrophage infiltration, cholesterol crystals, layered plaque, or calcium nodules within the lesion. Plaque characterization was performed for each IVOCT image frame. Table 1 provides a summary of the IVOCT features used in this study for predicting CV death.

### 2.4. Clinical Endpoint

Details regarding the occurrence of CV death were collected from the electronic medical records of University Hospitals Cleveland Medical Center in Cleveland, OH, USA. CV death was defined as death resulting from acute myocardial infarction, heart failure, cardiac shock, or other cardiovascular causes.

### 2.5. Statistical Analysis

We conducted various analyses on the IVOCT plaque features. Continuous features were presented as mean ± standard deviation, while categorical features were reported as frequencies. Statistical comparisons between the CV death and no-CV death groups were performed using a student t-test for continuous variables and the Chi-Square test for categorical variables. To assess the inter-correlations of IVOCT features, a heatmap analysis was conducted using the non-parametric Spearman’s rank correlation coefficient and hierarchical clustering. For the prediction of CV death, both univariate and multivariate logistic regressions were employed, with 95% confidence intervals (CI) calculated. In the multivariate logistic regression, features that showed significance (*p* < 0.05) in the univariate analysis were included. The discriminatory power of the models for predicting CV death was evaluated using the area under the receiver operating characteristic curve (AUC). The optimal cutoff values on the receiver operating characteristic (ROC) curve were determined based on the maximum sum of sensitivity and specificity. Statistical significance was defined as a *p*-value less than 0.05. All analyses were performed using R Studio software (version 1.4.1717, R Foundation for Statistical Computing, Vienna, Austria).

## 3. Results

This study included 104 patients with coronary artery disease who underwent IVOCT-guided PCI. No patients were excluded based on clinical characteristics. Among the 104 patients, the mean age was 67.1 ± 12.0 years, with 74 males (71.2%). During an average follow-up period of 19 months, CV death occurred in 24 patients (23.1%). Among the study population, 99 patients (95.2%) had hypertension, 53 patients (51.0%) were current smokers, and 56 patients (53.8%) had diabetes mellitus. The baseline characteristics of the study population are presented in Table 2.

We conducted a comparative analysis of plaque features between the CV death and no-CV death groups. The CV death group exhibited significantly larger lesion length, maximum calcium angle, and maximum calcium thickness compared to the no-CV death group (*p* < 0.05 for all) (Table 3). Similarly, there was a significant association between the occurrence of CV death and increasing FC features, including maximum FC angle, maximum FC area, FC surface area, and FC burden, compared to the no-CV death group. Notably, maximum FC area-T and FC surface area-T demonstrated the smallest *p*-values (*p* < 0.000001). However, quantitative features related to lumen, calcium depth, and minimum FC thickness did not show significant differences between the CV death and no-CV death groups. Furthermore, all vulnerable plaque features, such as microchannel and cholesterol crystal, were more frequently observed in the CV death group compared to the no-CV death group. Table 3 provides a comprehensive comparison of IVOCT plaque features using a Student’s *t*-test.

To address the issue of correlated features, we conducted a hierarchical clustering analysis of IVOCT plaque features using the non-parametric Spearman’s rank correlation coefficient (Figure 2). Among all the IVOCT plaque features, we identified 12 features (minimum lumen diameter, average lumen diameter, maximum FC area 1/2/3, FC surface area 1/2/3, and FC burden 1/2/3/T) with Spearman’s correlation coefficients exceeding 0.9. Consequently, we reduced the total number of features extracted from the IVOCT images from 31 to 14, focusing only on features with a rho-value greater than 0.9. Notably, within the heatmap, the FC features exhibited the smallest *p*-values, indicating their potential significance in relation to CV death prediction.

To assess the discriminatory ability of the features in predicting CV death events, we performed univariate and multivariate logistic regression analyses. In the univariate regression analysis, several features, including lesion length, maximum calcium angle, maximum calcium thickness, maximum FC angle, maximum FC area-T, FC surface area-T, cholesterol crystal, and layered plaque, demonstrated significant associations with CV death (Table 4). On the other hand, minimum/average lumen area and FC thickness did not independently predict CV death. In the multivariate regression analysis, only FC surface area-T (OR 2.38, CI 0.98–5.83, *p* = 0.03) exhibited a strong association with the occurrence of CV death (Table 4). The IVOCT plaque features that showed significant associations with CV death in both univariate and multivariate logistic regression analyses are summarized in Table 4.

Using the single best feature, identified as FC surface area-T from the multivariate regression analysis, we constructed an ROC curve to assess the predictive capability of the IVOCT plaque feature for CV death (Figure 3). FC surface area-1 made the smallest contribution, while FC surface area-2 (65 µm < T < 150 µm) contributed the most to CV death prediction. The combined AUC for FC surface area-T was 0.851 (95% CI 0.800–0.927, *p* = 0.0002). A significant difference in FC surface area-T was observed using a box-plot analysis (Figure 4). Additionally, Figure 5 presents a 3D visualization of high-risk and low-risk lesions in representative IVOCT pullbacks. The high-risk lesion exhibited thicker calcification (1.32 mm) and a larger FC surface area (39.10 mm^2^), whereas the low-risk lesion had thinner calcium (0.72 mm) and a smaller FC surface area (1.63 mm^2^).

## 4. Discussion

Building upon our previous research utilizing IVOCT imaging [50,53,55,56,57,58,59], we aimed to establish correlations between IVOCT plaque features and the occurrence of CV death. This study offers several noteworthy contributions. First, we employed our interactive OCTOPUS software to automatically compute plaque features such as FC thickness and FC surface area. Second, our analysis of features associated with CV death revealed the presence of high-risk features, including FC surface area, as well as unexpectedly low-risk features, such as TCFA thickness, when assessed in the context of stent-treated lesions. Third, preliminary ROC analysis indicates that features observed in IVOCT images hold the potential for predicting future adverse events, offering valuable insights into patient-management strategies.

The predictive power of FC surface area surpassed that of FC thickness, which has been commonly studied as a risk factor for lesion rupture in previous reports [41,60,61]. Our findings highlight that lesion area plays a more critical role in predicting CV death. This may be partially attributed to temporal trends in plaque pathogenesis. Typically, the formation of a plaque is preceded by the accumulation of lipid-laden macrophages, which later undergo apoptosis, resulting in the development of a necrotic lipid-rich core. Subsequently, fibrous tissue forms in the intima, leading to the formation of the FC. Thinning of the FC typically occurs after plaque enlargement [62]. Therefore, the “3D” lesion size may serve as an earlier indicator of plaque instability compared to the later appearance of a localized thin FC. Considering the diffuse inflammatory nature of atherosclerosis, it is crucial to analyze the entire 3D lesion, specifically focusing on the FC surface area. For this reason, the FC surface area was selected as a significant predictor for CV death, as it encompasses all relevant FC information, even though both FC area and FC angle were significant in univariate analysis. In another study, we also observed that FC surface area was a strong indicator for predicting the development of neo-atherosclerosis [63]. In contrast, assessments of the thin cap were not strong predictors of CV death. Specifically, neither minimum thickness nor FC surface area-1 was highly predictive. Nevertheless, it is important to note that the significance of FC thickness should not be overlooked. In our analysis, FC thickness was measured from a lesion that was subsequently treated with a stent, providing additional protection to the lesion. CV death is unlikely to be associated with the treated lesion, although in-stent thrombosis and restenosis are possible. It is likely that our assessments are associated with the extent of the disease. An important consideration is that our datasets show relatively high rates of stent under-expansion, which may explain the relatively high rates of cardiovascular death (23.1%). While a thin FC may signify local plaque instability, a larger surface area may indicate a more widespread burden of atherosclerotic disease. This could potentially translate into acute events occurring in regions outside the imaged plaque, perhaps even in another vessel.

Interestingly, luminal stenosis (minimum lumen area) was not a strong predictor of the risk of CV death. That is, the severity of a treated, flow-limiting stenosis is not a reliable indicator of future adverse events. Based on the arguments presented above, it does not appear to be a predictor of CV death-related disease in the rest of the heart. This is expected, as a more comprehensive 3D analysis of the vessel provides a more reliable assessment for CV death, considering numerous variable interactions, unlike a single measurement like the minimum lumen area. This finding aligns with a study conducted by Kim et al. [48], which found no significant differences in minimal lumen diameter and percentage of stenosis between lesions with and without adverse clinical outcomes in patients who underwent IVOCT-guided PCI, further supporting our results.

Long-term outcome-prediction studies have predominantly utilized CCTA and cardiac magnetic resonance angiography (CMRA). These studies have focused on clinical characteristics and high-risk plaque features such as spotty calcification, low-attenuation plaque, positive remodeling, and the napkin-ring sign [64,65]. Recently, there has been a growing number of studies aiming to predict long-term outcomes using radiomic features in CCTA [5,7,8,9,10,11,12]. For instance, Oikonomou et al. calculated a total of 843 radiomic features (including shape-related, first-order, and texture features) in CCTA images and correlated them with the occurrence of MACE within 5 years of the CCTA scan [9]. They discovered a high-risk radiomic profile of pericoronary fat that may be associated with an elevated cardiac risk. Similarly, Kolossváry et al. analyzed 935 radiomic features, including first-order and textural features with different bin sizes, to identify invasive and radionuclide imaging markers of plaque vulnerability [5]. Their results demonstrated that the most informative radiomic features were able to identify attenuated plaque as observed by intravascular ultrasound (IVUS), TCFA as observed by IVOCT, and NaF18-positivity. However, non-invasive imaging modalities such as CCTA and CMRA only have a moderate correlation with the gold-standard intravascular imaging techniques (e.g., IVOCT) and provide limited information about the artery wall and microscopic features of atherosclerosis. IVOCT, with its nearly histological resolution (axial: 10 μm, lateral: 20–40 μm) and optical contrast, offers a comprehensive assessment of coronary arteries [16]. Specifically, it enables a unique evaluation of microscopic plaque components such as FC, macrophages, cholesterol crystals, and microchannels, in addition to macroscopic plaques including fibrous and calcified plaques. Although IVOCT provides a better representation of the state of atherosclerosis compared to non-invasive imaging modalities, no studies have quantitatively analyzed IVOCT plaque characteristics for predicting future adverse outcomes. In this study, we utilized OCTOPUS [50] to perform plaque characterization in IVOCT images and, for the first time, identified FC surface area as a significant determinant of long-term CV death outcomes. Our results are promising and have the potential to optimize treatment strategies and improve short- and long-term outcomes.

Improved and automated characterization of atherosclerosis in IVOCT holds the potential to enable personalized treatments. The advancements in preventive and cardioprotective therapeutics over the past decade, including P2Y12 antagonists, direct oral anticoagulants, proprotein convertase subtilisin/kexin 9 (PCSK9) inhibitors, icosapent ethyl, glucagon-like peptide 1 (GLP-1) agonists, and others, highlight the need for personalized medicine approaches that ensure appropriate treatment for individual patients in a cost-effective manner. The automated identification of high-risk vessels would provide an opportunity to guide the implementation of intensive therapies in clinical practice and enhance patient cohorts for testing the effectiveness of emerging novel therapeutics. Furthermore, accurate identification of high-risk lesions could inform potential revascularization strategies. For instance, in addition to treating stenosis, an extra stent could be added to seal a high-risk lesion. The assessment of plaque changes with precise registration has the potential to facilitate mechanistic studies in drug development [66]. Additionally, the identification of high-risk IVOCT characteristics could offer insights into other imaging modalities.

This study has some limitations that should be acknowledged. First, it was a retrospective study conducted at a single center, and the sample size was relatively small. This may limit the generalizability of the findings to a larger population. Particularly, the number of features used in the multivariate logistic analysis was relatively large compared to the number of events, which may not accurately represent the underlying associations. Second, the study focused specifically on patients undergoing PCI with available IVOCT data. It remains unclear whether the identified features would be applicable and relevant to a more diverse and broader population. Third, despite the promising segmentation performance, the OCTOPUS software still requires manual editing.

In conclusion, our study demonstrates that patients with CV death have distinct plaque characteristics in IVOCT images compared to those without CV death. Particularly, FC surface area showed a strong predictive value, while features related to cap thickness were less predictive despite their emphasis in the literature regarding lesion vulnerability. These findings have potential implications for patient management, allowing the identification of individuals at higher risk for future events. Furthermore, correlating the features identified in our study with those observed in other imaging modalities, such as CCTA, through multi-modality imaging studies could provide valuable insights.

## Figures and Tables

**Figure 1 bioengineering-11-00843-f001:**
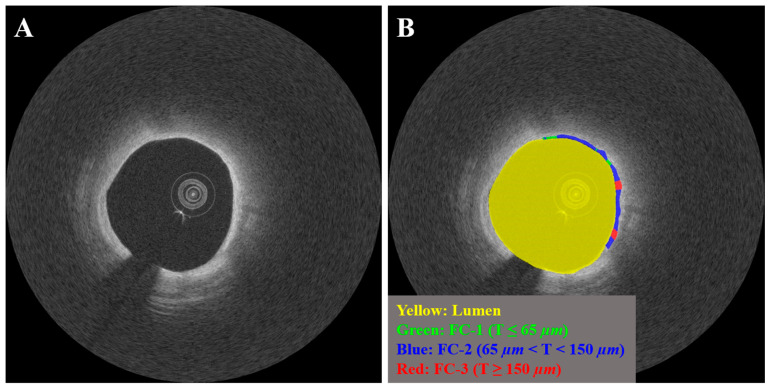
Representative IVOCT case with varying FC thicknesses (FC-1: T ≤ 65 µm, FC-2: 65 µm < T < 150 µm, and FC-3: T ≥ 65 µm). Panels include (**A**) Cartesian (*x*,*y*) IVOCT image and (**B**) IVOCT image overlaid with three FC classes. The lumen is represented in yellow, FC-1 in green, FC-2 in light blue, and FC-3 in red.

**Figure 2 bioengineering-11-00843-f002:**
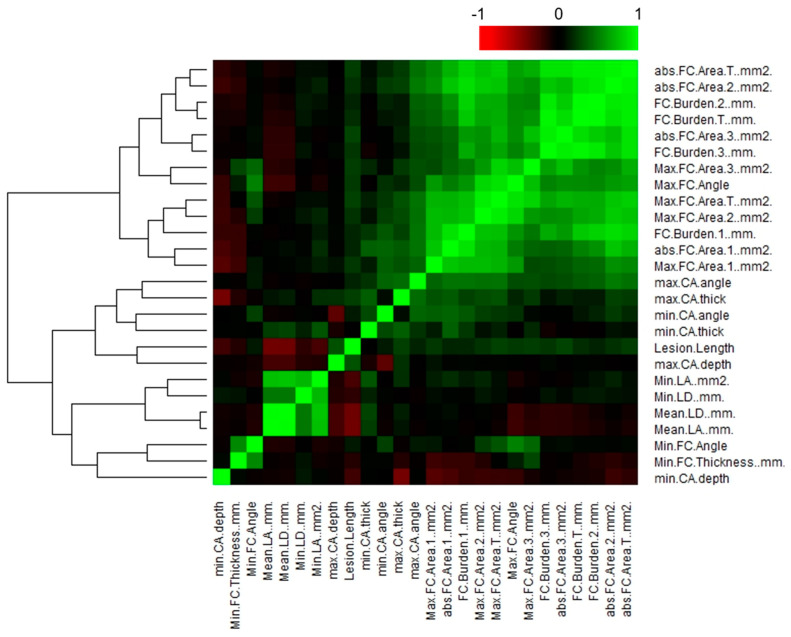
Correlation plot of IVOCT plaque features, illustrating hierarchical clustering and distinct clusters of feature correlation. The correlation coefficient (*R*) values are plotted against each other. The color key represents the explained variances: *R* values < 0.5 are shown in black, while greater values are depicted in green or red with increasing intensity. Using a Spearman correlation coefficient threshold of 0.9, a total of 12 features, including minimum/mean lumen diameter, maximum FC area-1/2/3, FC surface area-1/2/3, and FC burden-1/2/3/T, were excluded from further analysis.

**Figure 3 bioengineering-11-00843-f003:**
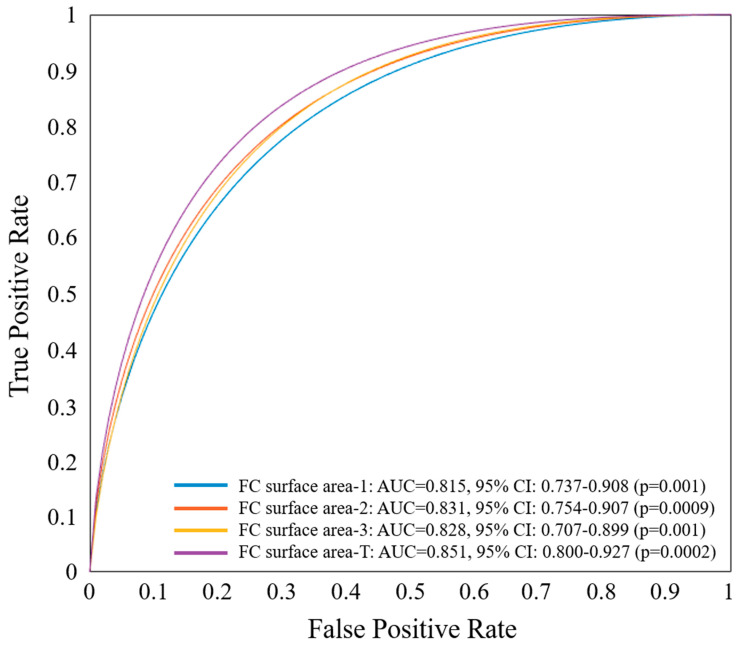
ROC curve analysis of the most significant IVOCT plaque feature, FC surface area-T, determined by multivariate logistic regression for predicting CV death. ROC curves of FC surface area 1–3 are also displayed. The FC area with T < 65 µm contributed the least (AUC: 0.815, 95% CI: 0.737–0.908, *p* = 0.001), while the FC area with 65 µm < T < 150 µm contributed the most (AUC: 0.831, 95% CI: 0.754–0.907, *p* = 0.0009) in predicting CV death. When combined, FC surface area-T had an AUC of 0.851 (95% CI 0.800–0.927, *p* = 0.0002).

**Figure 4 bioengineering-11-00843-f004:**
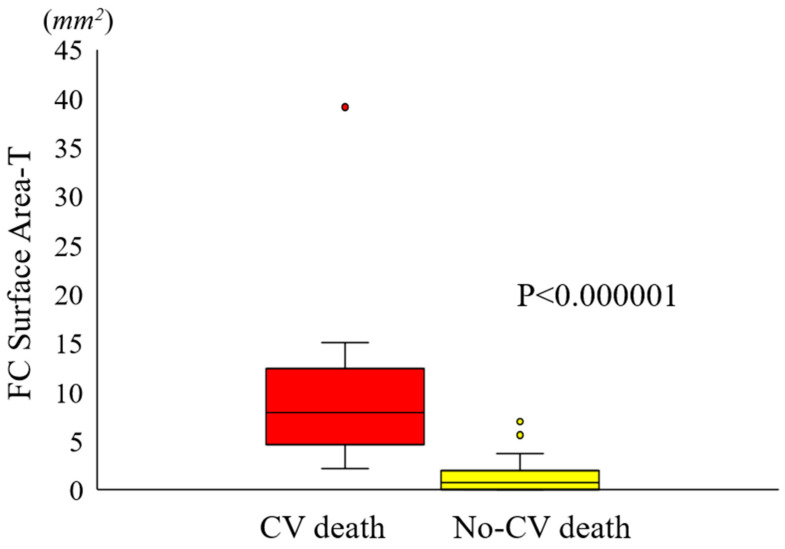
Box plot illustrating the disparity in FC surface area-T between the CV death and no-CV death groups. The FC surface area-T in the CV death group (9.63 ± 8.73 mm^2^) was significantly greater than that in the no-CV death group (1.37 ± 1.81 mm^2^) (*p* < 0.000001). The CV death group is represented in red, while the no-CV death group is depicted in yellow. The bars on the CV death and no-CV death groups indicate the respective standard deviations for each group.

**Figure 5 bioengineering-11-00843-f005:**
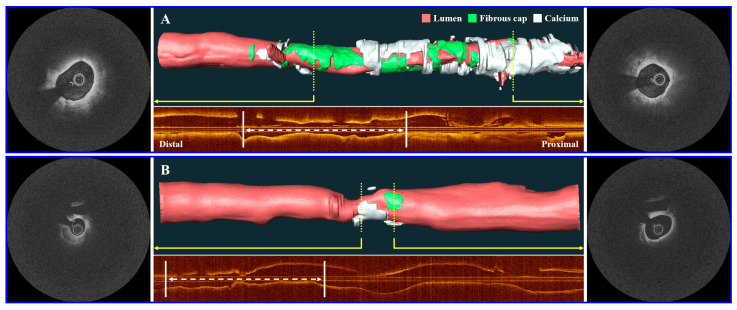
Three-dimensional (3D) visualization of high-risk and low-risk lesions on representative IVOCT pullbacks. The top panel (**A**) depicts a high-risk lesion with a maximum calcium thickness of 1.32 mm and an FC surface area of 39.10 mm^2^. The bottom panel (**B**) illustrates a low-risk lesion with a maximum calcium thickness of 0.72 mm and an FC surface area of 1.63 mm^2^. Subsequently, the high-risk case experienced a CV death, while the low-risk case remained event-free. The corresponding IVOCT image frames are displayed alongside the 3D vessel maps (indicated by yellow arrows). The corresponding longitudinal IVOCT maps are provided below each case, with white arrows indicating the location of the lesion where the stent was implanted. The lumen is represented in red, the FC in green, and the calcium in white.

**Table 1 bioengineering-11-00843-t001:** IVOCT plaque features, including lesion length and feature groupings (4 lumen, 6 calcium, 15 FC, and 5 vulnerable plaque features). FC refers to the fibrous cap, and VP represents vulnerable plaque.

N	Features	
1	Lesion length (mm)	
2	Lumen	Minimum lumen area (mm^2^)
3		Average lumen area (mm^2^)
4		Minimum lumen diameter (mm)
5		Average lumen diameter (mm)
6	Calcium	Maximum calcium angle (°)
7		Minimum calcium angle (°)
8		Maximum calcium thickness (mm)
9		Minimum calcium thickness (mm)
10		Maximum calcium depth (mm)
11		Minimum calcium depth (mm)
12	FC	Maximum FC angle (°)
13		Minimum FC angle (°)
14		Minimum FC thickness (mm)
15		Maximum FC area-1 (mm^2^)
16		Maximum FC area-2 (mm^2^)
17		Maximum FC area-3 (mm^2^)
18		Maximum FC area-T (mm^2^)
19		FC Surface area-1 (mm^2^)
20		FC Surface area-2 (mm^2^)
21		FC Surface area-3 (mm^2^)
22		FC Surface area-T (mm^2^)
23		FC burden-1
24		FC burden-2
25		FC burden-3
26		FC burden-T
27	VP	Microchannel
28		Macrophage Infiltration
29		Cholesterol Crystal
30		Layered Plaque
31		Calcium Nodule

**Table 2 bioengineering-11-00843-t002:** Baseline characteristics of the study population.

Characteristics	All (*n* = 104)	CV Death (*n* = 24)	No-CV Death (*n* = 80)	*p*-Value
Age (years)	67.1 ± 12.0	75.0 ± 8.5	72.0 ± 12.8	0.37
Male	74/104 (71.15%)	19/24 (79.2%)	52/80 (65.0%)	0.19
Physical Measurement				
Height (cm)	171.8 ± 9.8	173.6 ± 5.6	172.3 ± 12.3	0.67
Weight (kg)	93.7 ± 25.3	102.1 ± 36.4	91.1 ± 20.0	0.17
BMI (kg/m^2^)	31.73 ± 8.1	33.8 ± 12.0	30.8 ± 5.8	0.23
Medical History				
Hypertension	99/104 (95.2%)	24/24 (100.0%)	75/80 (93.8%)	0.21
Diabetes Mellitus	56/104 (53.8%)	13/24 (54.2%)	43/80 (53.8%)	0.97
Hyperlipidemia	90/104 (86.5%)	20/24 (83.3%)	70/80 (87.5%)	0.60
Previous PCI	8/104 (7.7%)	3/24 (12.5%)	5/80 (6.3%)	0.31
Previous Myocardial Infarction	60/104 (57.7%)	17/24 (70.8%)	43/80 (53.8%)	0.14
Heart Failure, LVEF < 30%	58/104 (55.8%)	16/24 (66.7%)	42/80 (52.5%)	0.22
Previous CABG	8/104 (7.7%)	2/24 (8.3%)	6/80 (7.5%)	0.89
Current Smoker (≤6 Months)	53/104 (51.0%)	15/24 (62.5%)	38/80 (47.5%)	0.20
Renal Dysfunction (Serum Creatinine > 2.0)	53/104 (51.0%)	16/24 (66.7%)	37/80 (46.3%)	0.08
Hemodialysis or Renal Transplant	12/104 (11.5%)	5/24 (20.8%)	7/80 (8.8%)	0.10
Pre-procedure Presentation				
STEMI/Cardiogenic shock	10/104 (9.6%)	2/24 (8.3%)	8/80 (10.0%)	0.81
NSTEMI/Unstable Angina	35/104 (33.7%)	5/24 (20.8%)	30/80 (37.5%)	0.13
Stable Angina	57/104 (54.8%)	13/24 (54.2%)	44/80 (55.0%)	0.94
Aortic stenosis	1/104 (1.0%)	1/24 (4.2%)	0/80 (0%)	0.07

**Table 3 bioengineering-11-00843-t003:** Comparison of IVOCT plaque features between CV death and no-CV death groups.

Features	CV Death (*n* = 24)	No-CV Death (*n* = 80)	*p*-Value
Lesion length (mm)	37.15 ± 14.15	28.56 ± 11.51	0.02
Maximum calcium angle (°)	245.19 ± 83.08	162.60 ± 71.08	0.0004
Minimum calcium angle (°)	20.31 ± 12.22	15.86 ± 5.24	0.06
Maximum calcium thickness (mm)	1.52 ± 0.24	1.24 ± 0.29	0.0007
Minimum calcium thickness (mm)	0.30 ± 0.10	0.27 ± 0.06	0.28
Maximum calcium depth (mm)	0.52 ± 0.18	0.50 ± 0.19	0.81
Minimum calcium depth (mm)	0.008 ± 0.010	0.014 ± 0.014	0.22
Minimum lumen area (mm^2^)	2.03 ± 1.04	2.07 ± 1.25	0.91
Average lumen area (mm^2^)	5.60 ± 2.37	5.87 ± 2.60	0.72
Minimum lumen diameter (mm)	0.96 ± 0.28	1.00 ± 0.37	0.75
Average lumen diameter (mm)	2.57 ± 0.56	2.62 ± 0.56	0.75
Maximum FC angle (°)	142.00 ± 51.21	61.40 ± 53.63	0.000003
Minimum FC angle (°)	29.81 ± 11.75	28.29 ± 27.06	0.83
Minimum FC thickness (mm)	0.0228 ± 0.0095	0.0409 ± 0.0567	0.21
Maximum FC area-1 (mm^2^)	0.48 ± 0.29	0.14 ± 0.21	0.000006
Maximum FC area-2 (mm^2^)	1.73 ± 0.75	0.62 ± 0.67	0.000001
Maximum FC area-3 (mm^2^)	1.38 ± 0.64	0.60 ± 0.65	0.0001
Maximum FC area-T (mm^2^)	3.60 ± 1.30	1.35 ± 1.23	0.00000009
FC Surface area-1 (mm^2^)	0.51 ± 0.49	0.07 ± 0.13	0.000002
FC Surface area-2 (mm^2^)	5.36 ± 5.53	0.72 ± 1.08	0.000002
FC Surface area-3 (mm^2^)	3.77 ± 3.76	0.58 ± 1.09	0.000006
FC Surface area-T (mm^2^)	9.63 ± 8.73	1.37 ± 1.81	0.0000002
FC burden-1	42.30 ± 42.10	5.56 ± 9.81	0.000002
FC burden-2	496.35 ± 796.34	53.59 ± 78.55	0.0007
FC burden-3	369.09 ± 570.24	40.31 ± 63.43	0.0004
FC burden-T	907.75 ± 1368.83	99.46 ± 121.49	0.0003
Microchannel	9 (37.5%)	11 (13.8%)	0.01
Macrophage Infiltration	21 (87.5%)	50 (62.5%)	0.02
Cholesterol Crystal	17 (70.8%)	12 (15.0%)	0.0000001
Layered Plaque	8 (33.3%)	3 (3.8%)	0.00004
Calcium Nodule	8 (33.3%)	8 (10.0%)	0.005

**Table 4 bioengineering-11-00843-t004:** Univariate/multivariate logistic regression for predicting CV death. Eight IVOCT plaque features showed significant correlation with CV death in the univariate logistic regression analysis. In the multivariate logistic regression analysis, only the FC surface area-T demonstrated a strong association with a higher prevalence of CV death (*p* < 0.05).

Features	Univariate Logistic Regression				Multivariate Logistic Regression			
	*p*-Value	Odd Ratio	Lower 95%	Upper 95%	*p*-Value	Odd Ratio	Lower 95%	Upper 95%
Lesion length (mm)	0.03	1.05	1.01	1.11	0.90	0.99	0.89	1.11
Maximum calcium angle (°)	0.002	1.00	1.01	1.01	0.29	1.01	0.99	1.02
Minimum calcium angle (°)	0.09	1.07	0.99	1.16				
Maximum calcium thickness (mm)	0.003	48.48	3.82	614.87	0.06	190.55	0.73	4993.57
Minimum calcium thickness (mm)	0.29	62.40	0.03	1224.3				
Maximum calcium depth (mm)	0.81	1.46	0.07	31.17				
Minimum calcium depth (mm)	0.07	0.02	0.00	1.30				
Minimum lumen area (mm^2^)	0.90	0.97	0.59	1.59				
Average lumen area (mm^2^)	0.71	0.96	0.75	1.21				
Maximum FC angle (°)	0.002	1.03	1.01	1.05	0.10	1.05	0.99	1.12
Minimum FC angle (°)	0.83	1.00	0.98	1.03				
Minimum FC thickness (mm)	0.23	0.00	0.00	920.14				
Maximum FC area-T (mm^2^)	0.0009	5.65	2.03	15.69	0.16	0.14	0.01	2.13
FC Surface area-T (mm^2^)	0.0002	2.08	1.42	3.04	0.03	2.38	0.98	5.83
Microchannel	0.10	3.00	0.82	10.98				
Macrophage infiltration	0.44	1.91	0.36	9.99				
Cholesterol crystal	0.0008	9.35	2.53	34.58	0.42	3.22	0.19	54.85
Layered plaque	0.01	9.09	1.55	53.39	0.48	5.18	0.06	489.67
Calcium nodule	0.09	3.36	0.82	13.78				

## Data Availability

The data presented in this study are available upon request from the corresponding author.

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
