# Peer review of "Plaque Characteristics Derived from Intravascular Optical Coherence Tomography That Predict Cardiovascular Death"

_bioengineering, 2024, doi:10.3390/bioengineering11080843_

Round 1

Reviewer 1 Report

Comments and Suggestions for Authors

In general, the paper sounds well in terms of Literature, methodology, experimental investigation, and conclusion of claim. However, I feel the authors didn't focus much their attention in comparing the results with bench-marking approach, which are discussed as part of literature. In this aspect, the manuscript requires revision. Also, Authors are asked to get the assistance of Native English Speaker to convert the minor flaws found in few places  of the manuscript.

Comments on the Quality of English Language

Authors are asked to get the assistance of Native English Speaker to convert the minor flaws found in few places  of the manuscript.

Author Response

Reviewer 1 1. In general, the paper sounds well in terms of Literature, methodology, experimental investigation, and conclusion of claim. However, I feel the authors didn't focus much their attention in comparing the results with bench-marking approach, which are discussed as part of literature. In this aspect, the manuscript requires revision.

(Response): Thank you for pointing this out. Comparing our results with bench-marking approaches could indeed be beneficial in demonstrating the robustness of our findings. However, this comparison is only feasible when the same data are used across all models. The primary objective of this study was to highlight the clinical importance of our novel FC surface area in predicting future adverse events. Additionally, since the bench-marking studies are thoroughly summarized in the Introduction, readers can easily compare our results with those studies. Therefore, we chose to focus on describing the features that were significantly associated with outcomes (e.g., FC surface area) or those that were not significantly associated (e.g., luminal stenosis) rather than indirectly comparing our results with previous studies. I hope this clarifies our approach for the reviewer. 2. Also, Authors are asked to get the assistance of Native English Speaker to convert the minor flaws found in few places of the manuscript. (Response): Thank you for pointing this out. We have had the manuscript revised by a native English speaker.

Reviewer 2 Report

Comments and Suggestions for Authors

A well written manuscript, clear, logical and easy to follow even for non-medical readers. Standard statistical analysis applied to existing data for new purposes. In my view, the current version is acceptable for publication as is. But if the authors can address the following minor points (a matter of adding a few words or sentences), the quality can be further improved.

While reading the manuscript, I had two lingering questions in my mind and looking for answers. One was the relatively small sample size. The other was the fact that the authors appeared to take the tomograms at face value, ignoring potential issues that are typical of tomographically obtained images, such as blurring, distortion and shadowing effects especially in regions of high contrast (and their features of interest happen to be in precisely such regions and often only a few pixels thick) and when the sample is in motion (presumably arteries in living patients can change in diameter while being imaged). In the Discussion section, the authors recognised that their sample size was relatively small. Perhaps the authors could also comment on the second question as well? This relates to their third point about segmentation and how its accuracy affects their analysis results and conclusions.

Another minor point: Abstract should be self-contained, thus full name for IVOCT and OCTOPUS should be given.

Author Response

Reviewer 2

A well written manuscript, clear, logical and easy to follow even for non-medical readers. Standard statistical analysis applied to existing data for new purposes. In my view, the current version is acceptable for publication as is. But if the authors can address the following minor points (a matter of adding a few words or sentences), the quality can be further improved.

  1. While reading the manuscript, I had two lingering questions in my mind and looking for answers. One was the relatively small sample size. The other was the fact that the authors appeared to take the tomograms at face value, ignoring potential issues that are typical of tomographically obtained images, such as blurring, distortion and shadowing effects especially in regions of high contrast (and their features of interest happen to be in precisely such regions and often only a few pixels thick) and when the sample is in motion (presumably arteries in living patients can change in diameter while being imaged). In the Discussion section, the authors recognised that their sample size was relatively small. Perhaps the authors could also comment on the second question as well? This relates to their third point about segmentation and how its accuracy affects their analysis results and conclusions.

(Response): Thank you for pointing this out. While we fully agree with the reviewer’s comments regarding issues such as blurring, distortion, and shadowing effects, we believe these are inherent limitations of OCT imaging in general, rather than specific limitations of our study. Additionally, based on our experience, the percentage of images affected by these issues is very small across the entire pullback, meaning their potential impact is likely to be minimal. For these reasons, we prefer not to include them as limitations of our study.

(Response): We agree that the segmentation accuracy of the OCTOPUS software could impact the analysis results. Therefore, we manually edited the automated results from OCTOPUS software if needed, as described in Section 2.2 (see below). This limitation has also been acknowledged in the Discussion section.

  • Section 2.2: ‘… If necessary, manual editing of the results was performed with an interactive editing tool, following the definitions provided in the ‘consensus document’ [54]. …’
  • Discussion: ‘… Third, despite the promising segmentation performance, the OCTOPUS software still requires manual editing.”

  1. Another minor point: Abstract should be self-contained, thus full name for IVOCT and OCTOPUS should be given.

(Response): Thank you for pointing this out. We have updated the Abstract accordingly.